# A fourth-generation high-dimensional neural network potential with accurate electrostatics including non-local charge transfer

Tsz Wai Ko [1✉], Jonas A. Finkler [2✉], Stefan Goedecker[2] & Jörg Behler [1]

Machine learning potentials have become an important tool for atomistic simulations in many fields, from chemistry via molecular biology to materials science. Most of the established methods, however, rely on local properties and are thus unable to take global changes in the electronic structure into account, which result from long-range charge transfer or different charge states. In this work we overcome this limitation by introducing a fourth-generation high-dimensional neural network potential that combines a charge equilibration scheme employing environment-dependent atomic electronegativities with accurate atomic energies. The method, which is able to correctly describe global charge distributions in arbitrary systems, yields much improved energies and substantially extends the applicability of modern machine learning potentials. This is demonstrated for a series of systems representing typical scenarios in chemistry and materials science that are incorrectly described by current methods, while the fourth-generation neural network potential is in excellent agreement with electronic structure calculations.

---

[1] Universität Göttingen, Institut für Physikalische Chemie, Theoretische Chemie, Tammannstraße 6, 37077 Göttingen, Germany. [2] Department of Physics, Universität Basel, Klingelbergstrasse 82, 4056 Basel, Switzerland. ✉email: tko@chemie.uni-goettingen.de; jonas.finkler@unibas.ch

Computer simulations nowadays have become an important tool in many fields of science like chemistry, molecular biology, physics, and materials science. The quality, and thus the predictive power, of the results obtained in these simulations crucially depends on the accurate description of the atomic interactions. While electronic structure methods like density functional theory (DFT) provide a reliable description of many types of systems, the high computational costs of DFT restrict its application in molecular dynamics (MD)[1] and Monte Carlo[2] simulations to a few hundred atoms preventing the investigation of many interesting phenomena. Larger systems can be studied by more efficient atomistic potentials, which avoid solving the electronic structure problem on-the-fly but instead provide a direct functional relation between the atomic positions and the potential energy. Atomistic potential energy surfaces (PESs) have been developed for many types of systems, and most of these potentials are based on physical approximations, which necessarily limit the accuracy of the obtained results.

With the advent of machine learning (ML) potentials[3–7] in recent year an alternative approach to the construction of PESs has emerged, which allows to combine the accuracy of quantum mechanical electronic structure calculations with the efficiency of simple empirical potentials. Many types of ML potentials have been proposed to date, like neural network potentials[8–12], Gaussian approximation potentials (GAPs)[13], moment tensor potentials (MTPs)[14], spectral neighbor analysis potentials (SNAPs)[15], and many others[16,17].

ML potentials can be classified into four different generations. Starting with the work of Doren and coworkers published in 1995[8], the first generation (1G) of ML potentials[18,19] has been applicable to low-dimensional systems depending on the positions of a few atoms only. This restriction has been overcome in high-dimensional neural network potentials (HDNNPs) proposed by Behler and Parrinello in 2007[9], which represented the first ML potential of the second generation (2G). In this generation, which employs the concept of nearsightedness[20], the total energy of the system is constructed as a sum of atomic energies, which depend on the local chemical environment up to a cutoff radius and —in case of HDNNPs—are computed by individual atomic neural networks. Most modern ML potentials making use of different ML algorithms, like HDNNPs, GAPs, MTPs, and SNAPs, belong to this second generation, and as standard methods for atomistic simulations they have been successfully applied to a wide range of systems.

A limitation of 2G ML potentials, which are applicable to tens of thousands of atoms, is the neglect of long-range interactions, i.e., electrostatics beyond the cutoff radius, but also dispersion interactions, which may substantially accumulate for condensed systems, are often truncated. This possible source of error, in particular for ionic systems, has been recognized early, and electrostatic corrections based on fixed charges have been proposed[13,21]. In more flexible third generation (3G) ML potentials, long-range electrostatic interactions are included by constructing environment-dependent atomic charges, which in case of 3G-HDNNPs are expressed by a second set of atomic neural networks[22,23]. These charges can then be used in standard algorithms like the Ewald sum to compute the full long-range electrostatic energy. Owing to the additional effort in constructing and using 3G ML potentials, most applications have been reported for molecular systems[12,24,25], while in simulations of condensed systems they are rarely used, as often long-range electrostatic interactions are efficiently screened.

A remaining limitation of 3G ML potentials is their inability to describe long-range charge transfer and different charge states of a system, since the atomic partial charges are expressed as a function of the local chemical environment only. Neglecting non-local charge transfer and changes in the global charge

distribution, which can be important in many systems[26,27], can result in qualitative failures as illustrated in Fig. 1 for the molecular model system $XC_7H_7O$ displayed in panel a. Depending on the choice of the functional group X in b, like an amino group $NH_2$ or its protonated form $NH_3^+$, different partial charges, which we use in this work as a qualitative fingerprint of the electronic structure, are obtained as shown in the plots of the DFT Hirshfeld charges on the right hand side. In particular the charge of the right oxygen atom depends on the choice of X, although X is far outside its local atomic environment displayed as dashed circle. As a consequence, ML potentials relying on a local description, like 2G- and 3G-HDNNPs, cannot distinguish these systems and the same charge is assigned to the right oxygen in both molecules, which is chemically incorrect. A second case is illustrated in Fig. 1c. In this case the OH group on the left is deprotonated resulting in a negative ion with two oxygen atoms almost equally sharing the negative charge. This charge is very different from the charge in the carbonyl oxygen of the neutral molecule. Still, again, the local environment of the carbonyl oxygen atom is identical, which is why 2G and 3G ML potentials cannot be applied to multiple charge states.

This limitation of local atomistic potentials in the description of long-range charge transfer and of systems in different charge

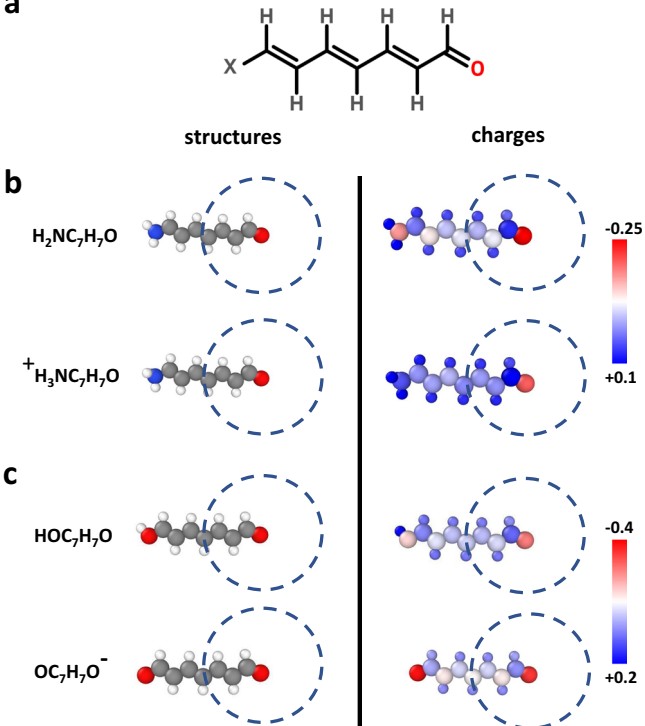

**Fig. 1 Illustration of long-range charge transfer in a molecular system.** In **a** the investigated molecule $XC_7H_7O$ with X representing different functional groups is shown. **b** The protonation of $NH_2$ group yields a positive ion and result in different charges of the oxygen atom as can be seen in the plot of the DFT atomic partial charges on the right side. In both cases, the local chemical environments of the oxygen atoms are identical within the cutoff spheres shown as dashed circles. **c** The deprotonation of the OH group yields a negative ion and both oxygen atoms become chemically equivalent with the nearly same negative partial charge. Also in this case the chemical environment of the right oxygen atom is identical to the neutral molecule although the charge distribution differs. All these cases cannot be correctly described by local methods like 2G and 3G ML potentials. The structure visualization for non-periodic systems was carried out using Ovito[66].

states has been recognized already some time ago, and for simple empirical force fields different solutions have been proposed[28–31]. In the context of ML potentials the first method that has been proposed to address this problem is the charge equilibration via neural network technique (CENT)[32–34]. In this method, a charge equilibration[28] scheme is applied, which allows for a global redistribution of the charge over the full system to minimize a charge-dependent total energy expression. The charges are based on atomic electronegativities, which are determined as a function of the local chemical environment and expressed by atomic neural networks similar to the charges in 3G-HDNNPs. This method has enabled the inclusion of long-range charge transfer in a ML framework for the first time, but due to the employed energy expression this method is primarily applicable to ionic systems[35–37], and the overall accuracy is still lower than in case of other state-of-the-art ML potentials. Recently, another promising method has been proposed by Xie, Persson and Small[38] aiming for a correct description of systems with different charge states. In this method, atomic neural networks are used that do not only depend on the local structure but also on atomic populations, which are determined in a self-consistent process. The training data for different populations has been generated using constrained DFT calculations, and a first application for $Li_nH_n$ clusters has been reported. Furthermore, an extension of the AIMNet method has been proposed[39], which can be used to predict energies and atomic charges for systems with non-zero total charge. Here, the interaction range between atoms is increased through iterative updates during which information is passed between nearby atoms. Although the resulting charges are not used to calculate explicit Coulomb interactions, many related quantities, such as electronegativities, ionization potentials or condensed Fukui functions can be derived.

In the present work, we propose a general solution for the limitations of current ML potentials by introducing a fourth-generation (4G) HDNNP, which is applicable to long-range charge transfer and multiple charge states. It consists of highly accurate short-range atomic energies similar to those used in 2G-HDNNPs and charges determined from a charge equilibration method relying on electronegativities in the spirit of the CENT approach. Both, the short-range atomic energies as well as the electronegativities are expressed by atomic neural networks as a function of the chemical environments. The capabilities of the method are illustrated for a series of model systems showcasing typical scenarios in chemistry and materials science that cannot be correctly described by conventional ML potentials. For all these systems we demonstrate that 4G-HDNNPs trained to DFT data are able to provide reliable energies, forces and charges in excellent agreement with electronic structure calculations. In the beginning of the following section the methodology of 4G-HDNNPs is introduced and the relation to other generations of HDNNPs and the CENT method is discussed. After that the results for a series of periodic and non-periodic benchmark systems are presented, including a detailed comparison to the performance of 2G- and 3G-HDNNPs. We show that previous generations of HDNNPs, which are unable to take distant structural changes into account, yield inaccurate energies and forces, and even distinct local minima of the PES can be missed, which are correctly resolved by the 4G-HDNNP. These results are general and equally apply to other types of 2G ML potentials.

## Results

**4G-HDNNP**. The overall structure of the 4G-HDNNP is shown schematically in Fig. 2 for an arbitrary binary system. Like in 3G-HDNNPs the total energy consists of a short-range part, which, as we will see below, requires in addition non-local information, and

an electrostatic long-range part, which is not truncated,

$$E_{total}(\mathbf{R}, \mathbf{Q}) = E_{elec}(\mathbf{R}, \mathbf{Q}) + E_{short}(\mathbf{R}, \mathbf{Q}). \quad (1)$$

The electrostatic part $E_{elec}(\mathbf{R}, \mathbf{Q})$ depends on a set of atomic charges $\mathbf{Q} = \{Q_i\}$, which are trained to reference charges obtained in DFT calculations, and the positions of the atoms $\mathbf{R} = \{\mathbf{R}_i\}$. An important difference to 3G-HDNNPs is that these charges are not directly expressed by atomic neural networks as a function of the local atomic environments, but they are obtained indirectly from a charge equilibration scheme based on atomic electronegativities $\{\chi_i\}$ that are adjusted to yield charges in agreement with the DFT reference charges, which here we choose to be Hirshfeld charges[40], but many choices are in principle possible.

Like in the CENT approach the atomic electronegativities are local properties defined as a function of the atomic environments using atomic neural networks. As in 2G- and 3G-HDNNPs there is one type of atomic neural network with a fixed architecture per element in the system making all atoms of the same type chemically equivalent, while the specific values of the electronegativities depend on the positions of all neighboring atoms inside a cutoff sphere of radius $R_c$. The positions of the neighboring atoms inside this sphere are specified by a vector $\mathbf{G}_i$ of atom-centered symmetry functions[41], which ensures the translational, rotational and permutational invariance of the electronegativities.

To predict the atomic charges, which are represented by Gaussian charge densities of width $\sigma_i$ taken from the covalent radii of the respective elements, a charge equilibration scheme[42] is used. In this scheme, the charge is distributed among the atoms in an optimal way to minimize the energy expression

$$E_{Qeq} = E_{elec} + \sum_{i=1}^{N_{at}} (\chi_i Q_i + \frac{1}{2} J_i Q_i^2) \quad , \quad (2)$$

with $E_{elec}$ being the electrostatic energy of the Gaussian charges and $J_i$ the element-specific hardness. The $J_i$ do not depend on the chemical environment and are constant for each element. While they are manually chosen in the CENT method, we optimize them during training. They are hence treated as free parameters like the weights and biases of the neural networks. For the electrostatic energy we then obtain

$$E_{elec} = \sum_{i=1}^{N_{at}} \sum_{j}^{N_{at}} \frac{erf\left(\frac{r_{ij}}{\sqrt{2}\gamma_{ij}}\right)}{r_{ij}} Q_i Q_j + \sum_{i=1}^{N_{at}} \frac{Q_i^2}{2\sigma_i\sqrt{\pi}} \quad (3)$$

with

$$\gamma_{ij} = \sqrt{\sigma_i^2 + \sigma_j^2} \quad . \quad (4)$$

To solve this minimization problem the derivatives of $E_{Qeq}$ with respect to the charges $Q_i$ are calculated and set to zero,

$$\frac{\partial E_{Qeq}}{\partial Q_i} = 0, \forall i = 1, .., N_{at} \Rightarrow \sum_{j=1}^{N_{at}} A_{ij} Q_j + \chi_i = 0 \quad (5)$$

where the elements of the matrix $\mathbf{A}$ are given by

$$[\mathbf{A}]_{ij} = \begin{cases} J_i + \frac{1}{\sigma_i\sqrt{\pi}}, & \text{if } i = j \\ \frac{erf\left(\frac{r_{ij}}{\sqrt{2}\gamma_{ij}}\right)}{r_{ij}}, & \text{otherwise} \end{cases} \quad (6)$$

Considering the constraint that the sum of all charges must be equal to the total charge $Q_{tot}$ of the system, the following set of linear equations is solved by including this constraint via the

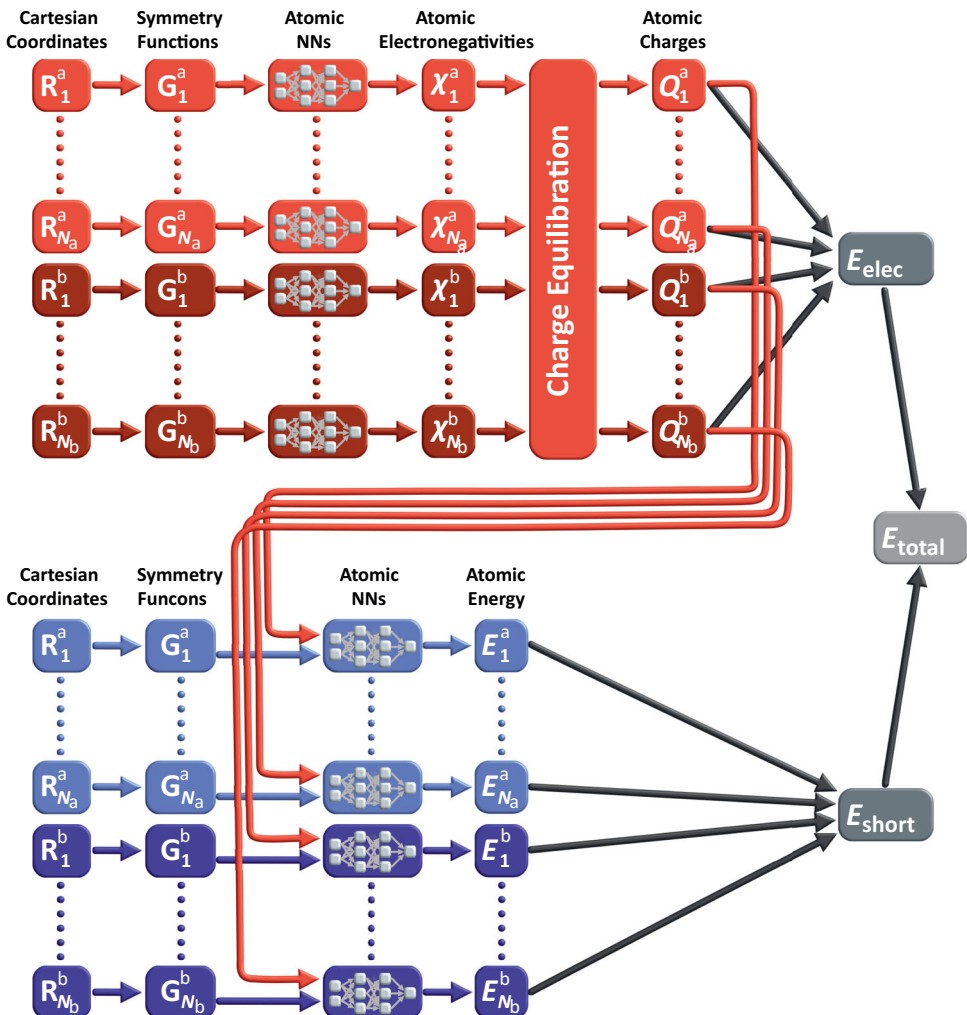

**Fig. 2 Schematic structure of a 4G-HDNNP for a binary system.** For a binary system containing $N_a$ atoms of element a and $N_b$ atoms of element b the total energy consists of a short-range energy $E_{short}$, which is a sum of atomic energies $E_i$, and a long-range electrostatic energy $E_{elec}$ computed from atomic charges $Q_i$. The atomic charges are determined by a charge equilibration method using environment-dependent atomic electronegativies $\chi_i$ expressed by atomic neural networks (red). These charges are then used to calculate the electrostatic energy and in addition serve as non-local input for the short-range atomic neural networks (blue) yielding the $E_i$. The geometric atomic environments are described by atom-centered symmetry function vectors $\mathbf{G}_i$, which depend on the Cartesian coordinates $\mathbf{R}_i$ of the atoms and serve as inputs for the atomic neural networks.

Lagrange multipliers.

$$\left( \begin{array}{c|c} \mathbf{A} & \begin{matrix} 1 \\ \vdots \\ 1 \end{matrix} \\ \hline 1 \ \dots \ 1 & 0 \end{array} \right) \left( \begin{matrix} Q_1 \\ \vdots \\ Q_{N_{at}} \\ \hline \lambda \end{matrix} \right) = \left( \begin{matrix} -\chi_1 \\ \vdots \\ -\chi_{N_{at}} \\ \hline Q_{tot} \end{matrix} \right) \tag{7}$$

Highly optimized algorithms are available for systems of linear equations, which can be efficiently solved for small and medium-sized systems containing up to about ten thousand atoms in a few seconds on modern hardware. For larger systems the cubic scaling of the standard algorithms can pose a bottleneck. In that case one could resort to using iterative solvers for which the most expensive part of each iteration is a matrix vector multiplication involving the matrix $\mathbf{A}$. This corresponds to the evaluation of the electrostatic potential at each atoms position for which numerous low-complexity algorithms, such as fast multipole methods, are known. In this way it is possible to reduce the effort from cubic to nearly linear scaling providing access to very large systems.

Overall, this process is like in the CENT[32], but the main difference is in the training process. While in CENT only the error with respect to the DFT energies is minimized, the atomic charges obtained during the charge equilibration process serve merely as intermediate quantities, which do not have a strict physical meaning. In the 4G-HDNNP proposed in this work, the charges are trained directly to reproduce reference charges from DFT, which therefore are qualitatively meaningful although one should be aware that atomic partial charges are not physical observables and different partitioning schemes can yield different numerical values[43].

Once the atomic electronegativities have been learned, a functional relation between the atomic structure and the atomic partial charges is available. The intermediate global charge equilibration step ensures that these charges depend on the atomic positions, chemical composition and total charge of the entire system, and thus in contrast to 3G-HDNNPs non-local charge transfer is naturally included.

In a second step, the local atomic energy contributions yielding the short-range energy according to

$$E_{short} = \sum_{i=1}^{N_{at}} E_i \tag{8}$$

have to be determined. Like in 2G-HDNNPs the short-range atomic energies are provided by individual atomic neural networks based on information about the chemical environments. An important difference to 2G-HDNNPs is that the atomic energies in addition depend on non-local information that is provided to the short-range atomic neural networks by using not only the atom-centered symmetry function values describing the positions of the neighboring atoms inside the cutoff spheres, but also the atomic partial charges determined in the first step (s. Fig. 2). This information is required to take into account changes in the local electronic structure resulting from possible long-range charge transfer, which has an immediate effect on the local many-body interactions.

The short-range atomic neural networks are then trained to express the remaining part of the total energy $E_{ref}$ according to

$$E_{short} = E_{ref} - E_{elec} = \sum_{i=1}^{N_{at}} E_i(\{\mathbf{G}_i\}, Q_i) \quad , \qquad (9)$$

where the electrostatic energy is determined based on the partial charges resulting from the fitted atomic electronegativities. Thus, by construction the goal of the short-range part is to represent all energy contributions that are not covered by the electrostatic energy such that double counting is avoided. In addition to the energies, also the forces are used for determining the parameters of the short-range atomic neural networks. We note that since the short-range energy depends on the atomic charges, which in turn are functions of all atomic coordinates, the derivatives $\partial E_{short}/\partial Q_i$ as well as $\partial Q_i/\partial \mathbf{R}$ have to be considered in the computation of the forces. Details on how these contributions can be efficiently computed, as well as many other details of the 4G-HDNNP method, can be found in the supplementary methods.

In summary, in contrast to the CENT method, the short-range interactions are not described through the charges resulting from the charge equilibration process but are described by separate short-range neural networks, which enables a more accurate description of the total energy.

**Overview of test systems.** In the following subsections we demonstrate the limitations of ML potentials based on local properties only and show how they can be overcome by the 4G-HDNNP. For this purpose we use a set of non-periodic and periodic systems, which cover a wide range of typical situations in chemistry and materials science. The non-periodic systems consist of a covalent organic molecule, a small metal cluster and a cluster of an ionic material covering very different types of atomic interactions. These examples demonstrate the simultaneous applicability of a single 4G-HDNNP to systems of different total charges and the correct description of long-range charge transfer and the associated electrostatic energy. As a periodic system we have chosen a small gold cluster adsorbed on a MgO(001) slab, which is a prototypical example for heterogeneous catalysis. We show that in contrast to established ML potentials, the 4G-HDNNP is able to reproduce the change in adsorption geometry of the cluster if dopant atoms are introduced in the slab far away from the cluster. In all cases, the 4G-HDNNP PES is very close to the results obtained from DFT.

While in theses examples we do not explicitly investigate the transferability of the potentials to different systems, we expect that the 4G-HDNNP in general provides an improved transferability compared to 2G and 3G ML potentials due to the underlying physical description of the global charge distribution and the resulting electrostatic energy. This expectation is supported by the fact that even traditional charge equilibration schemes with constant electronegativities are known to work well across different systems[44]. Furthermore, for the related CENT

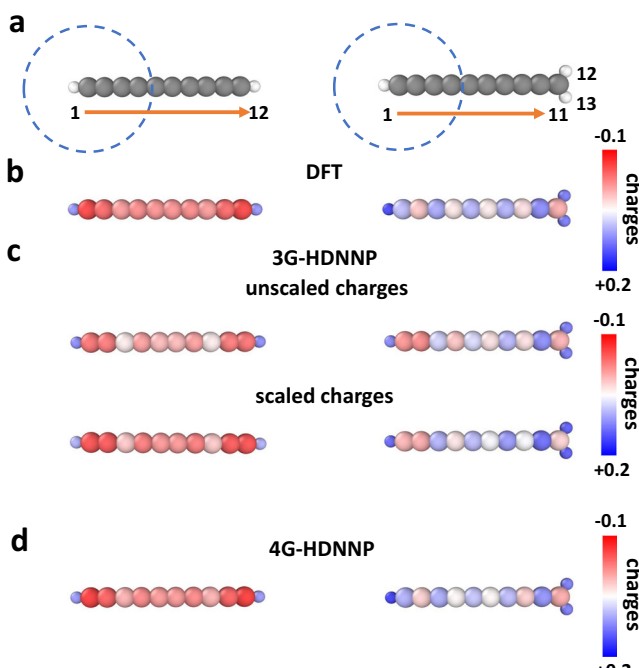

**Fig. 3 Charge redistribution in organic molecules. a** DFT-optimized structures of $C_{10}H_2$ (left) and $C_{10}H_3^+$ (right) with atom IDs. Carbon and hydrogen atoms are colored in gray and white, respectively. The dashed circle shows the cutoff radius of the left carbon atom defining its chemical environment. **b** shows the atomic partial charges obtained from DFT. The unscaled and scaled 3G-HDNNP charges are displayed in **c**, while the 4G-HDNNP charges are shown in **d**.

approach a broad transferability has already been demonstrated for different atomic environments[33].

**A benchmark for organic molecules.** The first model system we study is a linear organic molecule consisting of a chain of ten sp-hybridized carbon atoms terminated by two hydrogen atoms as shown in Fig. 3a. Molecules of this type have been studied before in electronic structure calculations[45–47]. For this molecule we will now demonstrate the applicability of 4G-HDNNPs to systems with long-range charge transfer induced by protonation, which changes the total charge and the local structure in a part of the system. Since the majority of existing machine learning potentials rely on local structural information only without explicit information about the global charge distribution and total charge, they are not simultaneously applicable to both neutral and charged systems.

This is different for 4G-HDNNPs, which naturally include the correct long-range electrostatic energy for any global charge present in the training set. Because of the protonation of the terminal carbon atom, its hybridization state changes to sp2 and the electronic structure of the resulting $C_{10}H_3^+$ cation is modified even at very large distances along the whole molecule, which is reflected in the differences of the DFT charges of the molecules in Fig. 3b, which have been structurally optimized by DFT. The geometries of both molecules are given in the supplementary tables.

Using a data set containing both molecules, we have constructed 2G-, 3G-, and 4G-HDNNPs using a cutoff radius $R_c = 4.23$ Å as illustrated by the circle in Fig. 3a for the example of the left carbon atom. In Fig. 3c we show the atomic partial charges obtained with the 3G-HDNNP in two forms: first as unscaled charges directly obtained from the atomic neural

network fits without any constraint for the correct total charge of the system, and second rescaled to ensure total charges of zero or one, respectively. It can be seen that the scaling process does not significantly improve the 3G-HDNNP charges.

The atoms in the left half of the molecule are far from the added proton such that their atomic environments differ only slightly due to the DFT geometry optimization. In addition, in the training set a lot of basically identical environments but different atomic charges are present for these atoms, which results in high fitting errors due to the contradictory information. As a consequence the neural networks assign averaged charges to these atoms, which differ qualitatively from the DFT reference charges of both systems. For instance, the 3G-HDNNP partial charges on atom 2, i.e., the left carbon atom, are almost identical in both molecules although they are very different in DFT. Note that the predicted charges of atoms 1–6 in $C_{10}H_2$ and $C_{10}H_3^+$ would be even exactly identical if the latter molecule would not have been relaxed after protonation. The charges obtained with the 4G-HDNNP shown in Fig. 3d, on the other hand, match the DFT charges very accurately for both molecules, as they can be distinguished in this method.

The inaccurate charges obtained with the 3G-HDNNP lead to a poor quality of the potential energy surface, and the same is observed for the short-range only 2G-HDNNP. In Table 1 we compare the errors of the total energies as well as the mean errors of the atomic charges and forces of all HDNNP generations for the DFT-optimized structures. It can be seen that the errors of all quantities obtained for the 4G-HDNNP are much lower than for the 2G- and 3G-HDNNPs. Further, we note that in several cases the energies obtained by the 3G-HDNNP are even worse than for the 2G-HDNNP, as the unphysical charge distribution to some extent prevents the accurate representation of the energy.

To investigate the forces in more detail, in Fig. 4 we plot the individual atomic forces in both molecules using the 2G-HDNNP and the 4G-HDNNP for the DFT-optimized structures. For all atoms in both molecules the 4G-HDNNP yields very low-force errors, with an average error of only 0.037 eV/Å underlining the quality of this PES. However, for the 2G-HDNNP the forces acting on the left half of $C_{10}H_3^+$ and on all atoms in $C_{10}H_2$ the force errors are significantly larger. The reason is again the 2G-HDNNP cannot distinguish both molecules for these atoms, and the force errors are only low close to the extra proton in $C_{10}H_3^+$, which can be recognized as a distinct local structural feature in the atomic environments of the right half of this molecule.

Interestingly, the relatively high errors of the 2G-HDNNP forces are not matched by high energy errors, which instead are

surprisingly low and smaller than 1 meV/atom for both molecules. This suggests that the total energy predicted by 2G-HDNNPs may benefit from error compensation in the atomic energies in that the atomic energies in the right half of $C_{10}H_3^+$ are adjusted to compensate the deficiencies of the atomic energies in the left half of the molecule.

**Metal clusters: Ag₃.** In this example, we investigate a small metal cluster, $Ag_3$, in two different charge states. The potential energy surface of small clusters is strongly influenced by the ionization state of the cluster and the ground state can differ as a function of the total charge of the cluster[48–51]. Owing to the small system size there are no long-range effects, and the full system is included in each atomic environment. Therefore, in principle 2G-HDNNPs should be perfectly suited to describe the PES of $Ag_3$, but this is only true as long as the total charge of the system does not change, since for a combination of data with different total charges, like $Ag_3^+$ and $Ag_3^-$, in the training set the unique relation between atomic positions and the energy is lost. The minimum-energy structures of both cluster ions obtained from DFT are shown in Fig. 5a along with the atomic partial charges. After training a 2G-HDNNP and a 4G-HDNNP to data containing both types of clusters, we have reoptimized the geometries by the respective HDNNP generation. As expected, the minima obtained

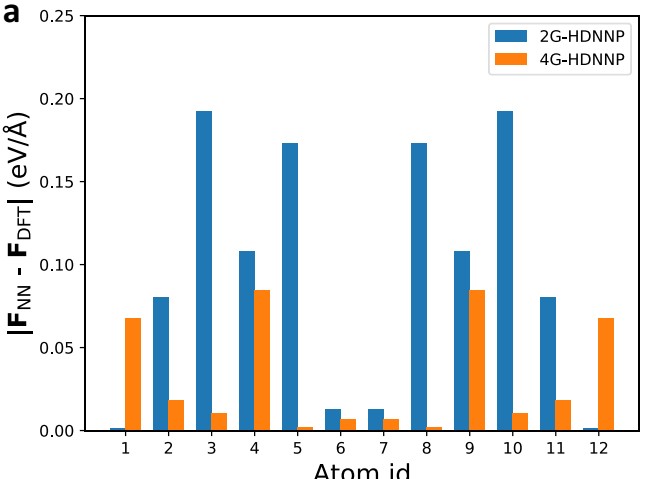

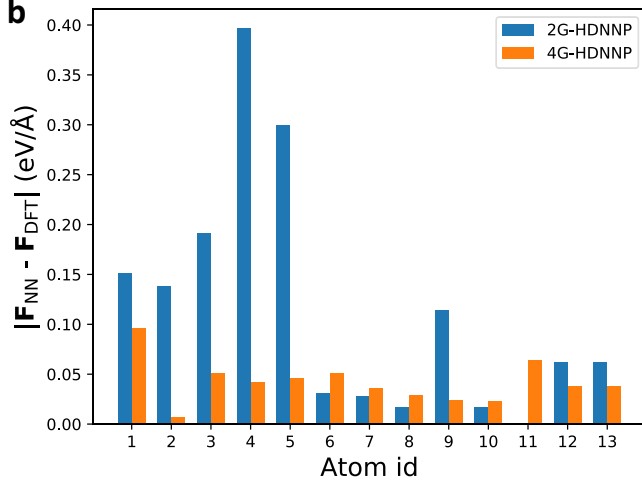

**Fig. 4 Force errors of the HDNNPs for the organic molecules.** 2G- and 4G-HDNNP forces for the atoms in the DFT-optimized structures of $C_{10}H_2$ and $C_{10}H_3^+$ (indicated in **a** and **b**, respectively).

**Table 1 Energy and charge error obtained for the organic molecules.** Energy error (meV/atom) and mean errors of the atomic charges ($10^{-3}$ e) and forces (eV/Å) of $C_{10}H_2$ and $C_{10}H_3^+$ with respect to DFT obtained with the different HDNNP generations for the DFT-optimized structures. For the 3G-HDNNP the results for scaled and unscaled charges are given.

| | | Energy | Charges | Forces |
|---|---|---|---|---|
| $C_{10}H_2$ | 2G-HDNNP | 0.684 | — | 0.095 |
| | 3G-HDNNP (unscaled) | 1.255 | 19.72 | 0.430 |
| | 3G-HDNNP (scaled) | 2.193 | 10.76 | 0.138 |
| | 4G-HDNNP | 0.463 | 4.820 | 0.032 |
| $C_{10}H_3^+$ | 2G-HDNNP | 0.922 | — | 0.127 |
| | 3G-HDNNP (unscaled) | 0.046 | 17.82 | 0.658 |
| | 3G-HDNNP (scaled) | 1.425 | 17.72 | 0.259 |
| | 4G-HDNNP | 0.176 | 5.048 | 0.042 |

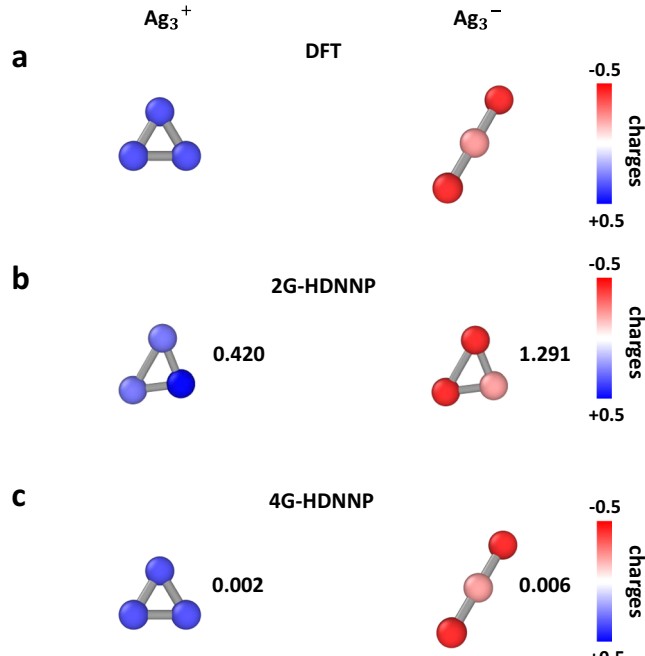

**Fig. 5 Optimized geometry and atomic charges of Ag clusters.** Structures and atomic partial charges of $Ag_3^+$ and $Ag_3^-$ optimized with DFT in **a**, the 2G-HDNNP in **b** and the 4G-HDNNP in **c**. The numbers give the root mean squared displacement (RMSD) in Å compared to the respective DFT minima. The partial charges in **b** are shown for illustration purposes only and have been obtained from a scaled 3G-HDNNP.

with the 2G-HDNNP (Fig. 5b) are identical for both charge states, but do not agree with any of the DFT structures. The 4G-HDNNP on the other hand, which in addition to the structural information also takes the total charge and the resulting partial charges into account, is able to predict the minima and also the atomic partial charges of both systems with very high accuracy (Fig. 5c). In this case, the inability of the 2G-HDNNP to distinguish between clusters is also apparent from the energy errors with respect to DFT. While the energy errors for $Ag_3^-$ and $Ag_3^+$ obtained from the 4G-HDNNP are only about 1.166 meV/atom and 0.320 meV/atom, respectively, the errors of the 2G-HDNNP are 0.605 and 2.017 eV/atom and thus several orders of magnitude larger. The 3G-HDNNP using scaled charges performs even worse and errors of 0.713 and 5.721 eV/atom are obtained. This is due to the non-physical electrostatic contribution calculated from the incorrectly predicted charges.

**NaCl cluster ions**. As the last non-periodic example we select a system with mainly ionic bonding, which is a positively charged $Na_9Cl_8^+$ cluster, and we analyze the changes of the PES, if a neutral sodium atom is removed. The initial structure of the cluster ion has been obtained from a DFT geometry optimization and is shown in Fig. 6. The sodium atoms are shown in purple, blue, and brown, while the chlorine atoms are displayed in gray. We then construct a second system by removing the brown sodium atom from the cluster while keeping the positions of the remaining atoms fixed. Since the overall positive charge of the cluster is maintained, the charge is redistributed throughout the new $Na_8Cl_8^+$ cluster ion.

To investigate the consequences of this change in the electronic structure on the PES, we compute and compare the energies and forces when moving the blue sodium atom along a one-dimensional path indicated by the arrow in Fig. 6 for both

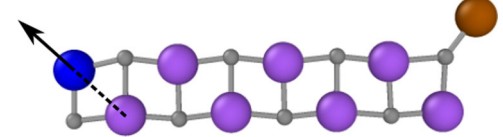

**Fig. 6 Optimized structure of the $Na_9Cl_8^+$ cluster.** Sodium atoms are shown in purple, blue and brown, chlorine atoms in gray. The arrow indicates the direction along which the blue sodium atom is moved for the energy and force plots in Fig. 7a and 7b. The position of this atom is defined by the Na–Na distance indicated as dashed line.

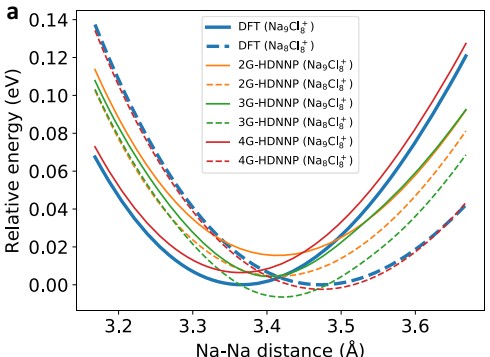

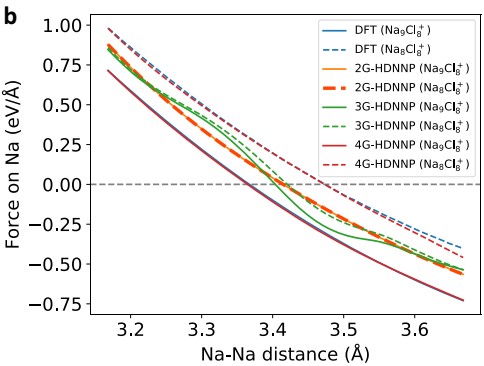

**Fig. 7 Relative energies and forces of the NaCl clusters. a** Relative energies of all potentials with respect to the DFT minima of the $Na_8Cl_8^+$ and the $Na_9Cl_8^+$ clusters as a function of the Na–Na distance and **b** forces acting on the blue sodium atom for the the path shown in Fig. 6. For the 3G-HDNNP unscaled charges have been used in this plot.

cluster ions. The distance to the closest neighboring sodium atom highlighted as dashed line is used to define the structure.

Figure 7 shows the energies for both systems obtained with DFT, as well as the 2G-, 3G- and 4G-HDNNPs. All energies are given as relative energies to the minimum DFT energy of the respective cluster ion and refer to the full systems. First, we note that the positions of the DFT minima differ by more than 0.1 Å, i.e., depending on the presence of the very distant brown atom the blue atom adopts different equilibrium positions. The 2G-HDNNP, however, is unable to distinguish these minima and instead the same local minimum Na–Na distance is found for both systems, which is approximately the average value of the two DFT minima. We note that the 2G-HDNNP energy curves of the two systems are not identical but there is an energy offset, as some of the atomic environments in the right part of the systems differ yielding different atomic energies. Since these environments do not change when moving the blue atom this offset is constant. For the 3G-HDNNP the same qualitative behavior is observed, and two very similar but not identical minima are found for both systems. Still, in case of the 3G-HDNNP the energy offset

between both systems is not merely a constant anymore, as the long-range electrostatic interactions between the blue and the brown atom in $Na_9Cl_8^+$ are position-dependent. We note that in spite of these qualitative differences with respect to DFT, the 2G- and 3G-HDNNP curves show only a deviation of about 1 meV per atom from the DFT curves. This is very small and in the typical order of magnitude of state-of-the-art ML potentials, and in the present case this apparently high accuracy hides the qualitatively wrong minima. Finally, the 4G-HDNNP energies for both systems are very accurate and the energy curves match the corresponding DFT curves very closely. Both distinct local minima are correctly identified and at the right positions.

Next, we turn to the forces shown in Fig. 7b. The results are fully consistent with our discussion of the energy curves. The DFT forces acting on the displaced atom are different for both cluster ions and well reproduced by the 4G-HDNNP. The 2G-HDNNP forces of both systems are exactly identical due to the constant offset between both energy curves (Fig. 7a), while 3G-HDNNP forces of both systems are slightly different due to the additionally included long-range electrostatics.

**$Au_2$ cluster on MgO(001).** As example for a periodic system we choose a diatomic gold cluster supported on the MgO(001) surface. Similar systems have attracted attention because of their catalytic properties for reactions like carbon monoxide oxidation, epoxidation of propylene, water-gas-shift reactions, and the hydrogenation of unsaturated hydrocarbons[52]. Theoretical[53,54] as well as experimental studies[55] have shown that the geometry of these clusters can be modified by the introduction of dopant atoms into the oxide substrate. This ability to control the cluster morphology is of great interest, as it can enhance the catalytic activity of the system[54]. 2G-HDNNPs have been used before to study the properties of supported metal clusters[56–58], but systems as complex as doped substrates to date have remained inaccessible, since long-range charge transfer between the dopant and the gold atoms is crucial to achieve a physically correct description of these systems.

For $Au_2$ at MgO(001) there are two main adsorption geometries, an upright "non-wetting" orientation of the dimer attached to a surface oxygen and parallel to the surface in a "wetting" configuration, in which the two Au atoms reside on two Mg atoms. DFT optimizations of the positions of the gold atoms with fixed substrate for the doped and undoped surfaces reveal that the presence of the dopant atoms changes the relative stability of both structures. On the pure MgO support (Fig. 8a) the minimum-energy structure is "non-wetting", while a flat "wetting" geometry is more stable if the MgO is doped by three aluminum atoms (Fig. 8b) corresponding to 2.86% of the slab. The Al dopant atoms were introduced into the 5th layer, resulting in a distance of >10 Å from the gold atoms. Despite this large separation, we found that by doping the charge on the $Au_2$ cluster is reduced (becomes more negative) by about 0.2 e compared to the same geometry for the undoped surface. This change in the electronic structure does not only lead to a switching in the energetic order of the geometries but also to a change of the bond-length between the gold atoms and the substrate.

The energy difference ($E_{wetting} - E_{non\text{-}wetting}$) between the wetting and non-wetting configurations calculated with different methods on a doped substrate are −2.7 meV for DFT, 375 meV for the 2G-HDNNP and −41 meV for the 4G-HDNNP. On an undoped substrate we obtained 929 meV for DFT, 375 meV for the 2G-HDNNP and 975 meV for the 4G-HDNNP. These numbers were obtained after the positions of the gold atoms were optimized. In case of the 2G-HDNNP, both optimizations yield the same structure. For the 2G-HDNNP the energy

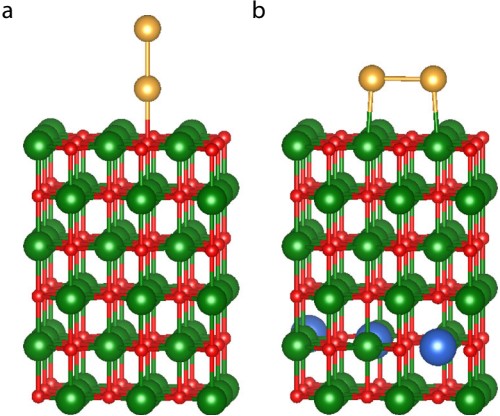

**Fig. 8 Geometry of $Au_2$ clusters on undoped and doped MgO(001) surface.** $Au_2$ cluster in the non-wetting geometry on the undoped **a** and the wetting geometry on Al-doped **b** MgO(001) surface represented by a periodic (3 × 3) supercell. Au atoms are shown in yellow, O in red, Mg in green and Al in blue. The configuration of the gold cluster has been optimized by DFT for a fixed substrate. The structure visualization for periodic systems was carried out using VESTA[67].

differences for the doped and undoped systems are exactly the same as the dopant atoms are outside the local chemical environments of the gold atoms. Thus, the 2G-HDNNP cannot take the change of the PES by doping into account. The DFT and 4G-HDNNP results agree in that there is a slight preference for the wetting configuration for the doped surface, while in the undoped case the non-wetting configuration is clearly more stable.

An analysis of the PES for the case of the non-wetting geometry for the doped and undoped slabs is given in Fig. 9, which shows the energies relative to the minimum DFT energies of the respective systems as a function of the distance between the bottom Au atom and its neighboring oxygen atom for DFT, the 2G-HDNNP and the 4G-HDNNP. The energy curves of the 4G-HDNNP and DFT are very similar and can resolve the different equilibrium bond lengths for the doped (4G-HDNNP: 2.342 Å; DFT: 2.332 Å) and undoped (4G-HDNNP: 2.177 Å; DFT: 2.190 Å) substrates. The 2G-HDNNP yields the same adsorption geometry with a bond-length of 2.256 Å in both cases, while the energies substantially differ from the DFT values with the main effect of the dopant being a constant energy shift between both substrates, similar to what we have observed in the presence or absence of the additional sodium atom in the NaCl cluster.

## Discussion

In this work, we developed a fourth-generation high-dimensional neural network potential with accurate long-range electrostatic interactions, which is able to take long-range charge transfer as well as multiple charge states of a system into account. The new method is thus applicable to chemical problems, which are incorrectly described by current machine learning potentials relying on a local description of the atomic environments only.

The 4G-HDNNP combines the advantages of the CENT approach and conventional high-dimensional neural network potentials of second and third generation by being generally applicable to all types of systems and providing a very high accuracy. Employing environment-dependent atomic electronegativities, which are expressed by atomic neural networks, a charge equilibration method is used to determine the global charge distribution in the system. The resulting charges are then used to compute the long-range electrostatic energy, as well as to include information about the global electronic structure into the

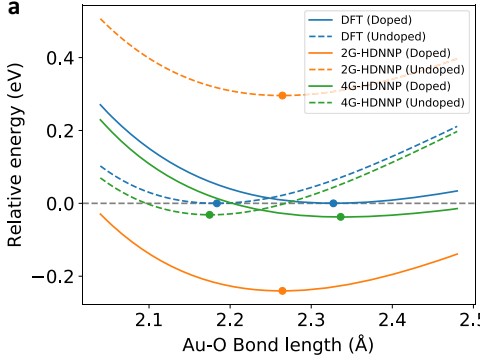

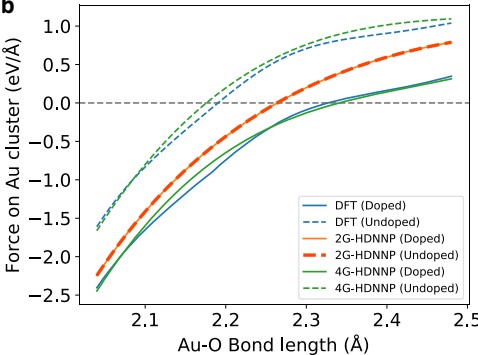

**Fig. 9 Energies and forces for the gold cluster. a** Relative energy and **b** sum of forces acting on the Au$_2$ cluster for the cluster adsorbed at the MgO(001) substrate for the non-wetting geometry for the Al-doped and undoped cases. The local minima of the energy curves are marked with a dot. The Au–O bond-length refers to the distance between the Au closest to the surface and its neighboring oxygen atom.

short-range atomic energy contributions represented by a second set of atomic neural networks.

The superiority of the 4G-HDNNP potential energy surface with respect to established 2G- and 3G-HDNNPs has been demonstrated for a series of systems, where conventional methods give qualitatively wrong results. In addition to the qualitatively correct description, we also obtained a clearly improved quantitative agreement of energies, forces and atomic charges with the underlying DFT data, and we could demonstrate that local minimum structures that are missed by the previous generations of HDNNPs are correctly identified by the new method.

The results obtained in this work are general and equally valid for other types of machine learning potentials relying on environment-dependent atomic energies only. Thus, the 4G-HDNNP is a vital step for the further development of next-generation ML potentials providing a correct description of the PES based a global charge distribution.

## Methods
**Neural network potentials.** The HDNNPs reported in this work have been constructed using the program RuNNer[59–61]. Atom-centered symmetry functions[41] have been used for the description of the atomic environments within a spatial cutoff radius set to 8–10 Bohr depending on the system. For a given system, the same parameters of the symmetry functions and the same atomic neural network architectures have been used for the different generations of HDNNPs being compared, and the parameters and cutoff radii for all systems can be found in supplementary tables. The functional forms of the symmetry functions are given in ref. [41]. In all examples, the atomic neural networks consist of an input layer with the number of symmetry functions ranging from 12 to 54 depending on the specific element and system, two hidden layers with 15 neurons each, and an output layer with one neuron providing either the atomic short-range energy or electronegativity. Forces have been obtained as analytic energy derivative. The activation

functions in the hidden layers and the output layer were the hyperbolic tangent and the linear function, respectively.

In all cases 90% of the available reference data was used for training the HDNNPs while the remaining 10% of the data points were used as an independent test set to confirm the reliability of PESs and detect possible over-fitting. Energies and forces were used for training the short-range atomic neural networks.

Moreover, a screening of the short-range Coulomb electrostatic interaction was applied in order to facilitate the fitting of the short-range energies and forces obtained from Eq. (9)[23]. The inner and outer cutoff radius for screening of the electrostatic interaction have been set to 1.69–2.54 Å and the cutoff of the symmetry functions, respectively. The widths of the Gaussian charge densities in Eq. (4) have been set to the covalent radii of the elements. All the details of the training process and the validation strategies for HDNNPs in general can be found in recent reviews[60,61].

The HDNNP-based geometry optimizations were performed using simple gradient descent algorithms and the numerical threshold of the forces was set to $10^{-4}$ Ha/Bohr $\approx 0.005$ eV/Å, which is the same convergence used in the DFT calculations used for validating the HDNNP results.

**DFT calculations.** The DFT reference data has been generated using the all-electron code FHI-aims[62] employing the Perdew–Burke–Ernzerhof[53] (PBE) exchange-correlation functional with light setting. The total energy, sum of eigenvalues, and charge density for all systems except Au$_2$-MgO were converged to $10^{-5}$ eV, $10^{-2}$ eV, and $10^{-4}$ e, respectively. For the Au$_2$-MgO systems stricter settings have been applied by multiplying each criterion by a factor 0.1 in combination with a $3 \times 3 \times 1$ k-point grid. Spin polarized calculations have been carried out for the Au$_2$-MgO, NaCl and Ag$_3$ systems. Reference atomic charges were calculated using Hirshfeld population analysis[40]. In principle any other charge partitioning scheme could be used in the same way.

The data set of the C$_{10}$H$_2$/C$_{10}$H$_3^+$ molecules and the Ag$_3$ clusters have been constructed by performing Born-Oppenheimer molecular dynamics[64] simulations for each system at 300 K with 5000 steps at a time step of 0.5 fs. A Nosé-Hoover thermostat[65] was applied to run simulations in the canonical (NVT) ensemble, and the effective mass was set to 1700 cm$^{-1}$. In addition, the trajectory path during the geometry relaxations up to a numerical convergence of 0.001 eV/Å of the forces was also added to the data set to have sufficient sampling close to equilibrium structures. The geometry optimization of the Ag$_3^-$ system has been terminated when reaching forces below 0.0015 eV/Å.

In case of the NaCl cluster and the Au$_2$ cluster at the MgO surface the reference data set consists of two structurally different types of systems, and half of the data set was dedicated to each of the two cases. We performed a random sampling along the trajectories depicted in Figs. 7 and 9 and added further Gaussian distributed displacements to ensure sufficient sampling of the PES in the vicinity of the structures of interest. For the NaCl cluster we used Gaussian displacements with a standard deviation of 0.05 Å. As in the Au$_2$-MgO system we only investigated the change in geometry of the Au$_2$ cluster, while the MgO substrate remained fixed during all geometry relaxations, we used a smaller magnitude of the Gaussian displacements for the substrate than for the cluster. A standard deviation of 0.02 Å was used for the substrate and 0.1 Å was used for the gold cluster. Half of the data set consists of structures with an undoped substrate, while the other half includes a doped substrate. Half of the samples of each substrate configuration were generated with the Au$_2$ cluster in its wetting configuration, and the other half with the cluster in its non-wetting configuration. The total number of reference data points for the NaCl cluster and Au$_2$-MgO slab is 5000, while the the Ag$_3$ clusters and the organic molecule it is 10,019 and 11,013, respectively.

## Data availability
The datasets used to train the NNPs presented in this paper have been published online[68]. All data that support the findings of this study are available in the Supplementary information file or from the corresponding author upon reasonable request.

## Code availability
All DFT calculations were performed using FHI-aims (version 171221_1). The HDNNPs have constructed using the program RuNNer, which is freely available under the GPL3 license at https://www.uni-goettingen.de/de/software/616512.html.

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

## Acknowledgements
We are grateful for the financial support from the Deutsche Forschungsgemeinschaft (DFG) (BE3264/13-1, project number 411538199) and the Swiss National Science Foundation (SNF) (project number 182877 and NCCR MARVEL). Calculations were performed in Göttingen (DFG INST186/1294-1 FUGG, project number 405832858), at sciCORE (http://scicore.unibas.ch/) scientific computing center at University of Basel and the Swiss National Supercomputer (CSCS) under project s963D/C03N05.

## Author contributions
Both research groups contributed equally to this project. J.B. and S.G. conceived the 4G-HDNNP approach and initiated the research project. T.W.K. and J.A.F. worked out the practical algorithms for the approach and implemented it in the RuNNer software written by J.B. All calculations were performed by T.W.K. and J.A.F. All authors contributed ideas to the project and jointly analyzed the results. T.W.K. and J.A.F. wrote the initial version of the manuscript and prepared the figures, all authors jointly edited the manuscript. T.W.K. and J.A.F. contributed equally to this paper.

## Funding

## Competing interests
The authors declare no competing interests.
