## [Peer Review File · Nature Communications]

REVIEWERS' COMMENTS

Reviewer #1 (Remarks to the Author):

The manuscript by Ko et al describes the development of a neural-network potential to incorporate long-range charge transfer. The work builds up on very active developments of machine learning models for molecular systems, and the ever-growing inclusion of physical laws. One piece that's critically missing in the field is the incorporation of long-range electrostatics, where "long" goes beyond the cutoff of the ML representation.

To accommodate for this, the authors describe an ML architecture that has two main aspects: a charge-equilibration scheme that's inherently long range, which feeds into a short-range predictor.

I find the approach compelling in that it robustly describes important aspects of short and long-range electrostatics. Unlike other recent ML developments, the physics is rather separated from the ML architecture: predictions are fed into physically motivated equations (like charge transfer) rather than building an ML architecture that satisfies this condition a priori.

All in all, I find the work technically extremely strong, but lack new insight. It doesn't bring about new ideas on ML architectures, but rather operates on a sophisticated sequential steps between ML predictions and physical equations.

Reviewer #2 (Remarks to the Author):

This manuscript reports a new neural network potential that explicitly includes contributions to the energy from interactions between atomic charges. The approach merges approaches developed previously by the authors: the CENT charge equilibration scheme of Goedecker (Ref. 36) and the second-generation (2-G) neural net potentials of Behler. The combination leads to the simple, but elegant, model of Figure 2. The upper neural network allows the electronegativity to vary with the environment of an atom. A global charge equilibration scheme then assigns charges to each atom. The total energy includes interactions between charges (Eq. 3). In addition, the charges are included, along with symmetry functions, as features to the neural network potential used predict the short-range interactions between atoms.

The manuscript does an excellent job of placing this work in the context of past work, by categorizing the methods via generations (1G through 4G), with 2G being the original form of Behler and Parrinello in Ref 9, and 4G being the current approach. 3G refers to methods that include atomic charges, but that determine these charges only from the local environment of the atom. The manuscript includes compelling examples (e.g. Fig 1) of where such 3G models will fail, and how the 4G model can overcome these failures by assuming that only the electronegativity is a function of the local environment.

The model is demonstrated on well-chosen model systems, including both non-periodic and periodic cases. The results provide strong evidence that the 4G model leads to improved performance and that this increase in performance is due to the inclusion of global charge equilibration.

A potential limitation of the paper is the extent to which transfer between systems is addressed. The data used here is from Molecular Dynamics trajectories, and the train test splits appear to randomly sample time points along these trajectories. (In some cases, structures are also included from random perturbations to these sampled structures.) Each test structure is thereby likely to be very close to chemical structures included in the training set. This is a fairly limited test of transfer. The results presented here therefore demonstrate that the 4G models provide accurate

functional forms for the potentials of specific molecules or periodic systems. For 2G models, past work has found good transfer between molecules within a fairly broad class of molecules. If the authors agree that such transfer between systems remains to be demonstrated for these 4G models, it may be useful to include a brief discussion of this in the paper.

The lack of a strong demonstration of transfer between systems may also limit the extent to which this paper is of interest to a broad audience. However, due to the simplicity and elegance of the model, and the clear demonstration that the 4G model can describe multiple charge states, I believe it is of sufficiently broad interest to warrant publication in Nature Communications.

The manuscript is very well written, but I do have two suggestions the authors may want to consider:

1) How are the hardness terms (J of Eq. 2) adjusted during training? In the CENT reference DOI: 10.1103/PhysRevB.92.045131, they are stated as being hand tuned. If these are also hand-tuned here, it may be helpful to include additional details on this process in the supporting information. If they are included in the training of the neural networks, it may be helpful to include this schematically in Fig. 2.

2) The results from the 3G model for the Ag clusters are not included in the main manuscript, other than to show the charges in a color scheme attached to the 2G results in Fig. 5. It may be helpful to include the 3G results in the sentence above Fig. 5 that compares the energy and forces of 4G to those of 2G. (In my first reading, I was distracted by the lack of comparison to 3G and why it was not included.)

I also noticed a few typos:

First paragraph below Fig. 2:
are no physical observables  are not physical observables

First sentence of section III
chapter  section

First full paragraph below Table 1:
word "the" repeated twice

Consider replacing the phrase "distinguish both clusters" with "distinguish between clusters"

Reviewer #3 (Remarks to the Author):

The authors propose an approach to include non-local information of charge populations into local machine learning models. The approach is elegant, as it uses local quantities predicted by HDNN potentials in combination with a global charge equilibration procedure in order to derive partial charges for an atomistic system of arbitrary charge. These charges are then in turn used to polarize the energy predictions of another set of HDNN potentials. This circumvents several problems local ML approaches have with describing the charge distributions of systems with differently charged states. The advantages of doing so are demonstrated clearly by the conducted experiments. Overall the paper is well written and I would recommend it for publication, provided a few minor issues are addressed.

The authors should discuss the computational effort of their approach and its implications for practical simulations. Since a system of equations needs to be solved during the charge equilibration, the method formally scales as $\sim O(N^3)$ compared to previous generations of the

potential which achieve a scaling of $\sim O(N)$. As such, one would expect that the maximum size of the system which can be treated or evaluation time during dynamics simulations is reduced.

Regarding the computation of the forces, it should be made clear whether the effect of the positions on the charges are considered in the required derivatives of the energy, as they are now environment dependent. This plays a role in the Coulomb expression for the energy, but also in the standard atomic networks, as these depend on the local charge. E.g.:

$$\frac{(dE(G(R),q(R)))/(dR_a)}{(dq(R))} = \frac{(dE(G(R),q(R)))/(dG(R))}{(dq(R))} \frac{(dG(R))}{(dR_a)} + \frac{(dE(G(R),q(R)))/(dq(R))}{(dR_a)}$$

This could for example influence the force error in Table I of the main text.

General comments:

There is a typo on page 9 in the sentence: "The Al dopant ...", where "then" is used instead of "than".

It would be helpful to include braces in the summation in Eq. 2 on page 3 to make clear, that both – the second and third term – are included in the sum over i .

ref 18 could be uploaded to arxiv so that this ref can be found by the reader.

1 Reviewer #1 (Remarks to the Author):

The manuscript by Ko et al describes the development of a neural-network potential to incorporate long-range charge transfer. The work builds up on very active developments of machine learning models for molecular systems, and the ever-growing inclusion of physical laws. One piece that's critically missing in the field is the incorporation of long-range electrostatics, where "long" goes beyond the cutoff of the ML representation.

To accommodate for this, the authors describe an ML architecture that has two main aspects: a charge-equilibration scheme that's inherently long range, which feeds into a short-range predictor.

I find the approach compelling in that it robustly describes important aspects of short and long-range electrostatics. Unlike other recent ML developments, the physics is rather separated from the ML architecture: predictions are fed into physically motivated equations (like charge transfer) rather than building an ML architecture that satisfies this condition a priori.

All in all, I find the work technically extremely strong, but lack new insight. It doesn't bring about new ideas on ML architectures, but rather operates on a sophisticated sequential steps between ML predictions and physical equations.

We thank the reviewer for this very positive assessment of our work. We would like to mention that the structure of our method shown in Fig. 2 represents a novel architecture that has not been used in any ML potential to date and now allows to combine the strengths of physical laws and machine learning. This new scheme provides access to physical insights for a wide range of systems which hitherto have been inaccessible by ML potentials. The focus of the present work is on the introduction of the method and the demonstration of its capabilities

using representative systems for different application scenarios.

2 Reviewer #2 (Remarks to the Author):

This manuscript reports a new neural network potential that explicitly includes contributions to the energy from interactions between atomic charges. The approach merges approaches developed previously by the authors: the CENT charge equilibration scheme of Goedecker (Ref. 36) and the second-generation (2-G) neural net potentials of Behler. The combination leads to the simple, but elegant, model of Figure 2. The upper neural network allows the electronegativity to vary with the environment of an atom. A global charge equilibration scheme then assigns charges to each atom. The total energy includes interactions between charges (Eq. 3). In addition, the charges are included, along with symmetry functions, as features to the neural network potential used predict the short-range interactions between atoms.

The manuscript does an excellent job of placing this work in the context of past work, by categorizing the methods via generations (1G through 4G), with 2G being the original form of Behler and Parrinello in Ref 9, and 4G being the current approach. 3G refers to methods that include atomic charges, but that determine these charges only from the local environment of the atom. The manuscript includes compelling examples (e.g. Fig 1) of where such 3G models will fail, and how the 4G model can overcome these failures by assuming that only the electronegativity is a function of the local environment.

The model is demonstrated on well-chosen model systems, including both non-periodic and periodic cases. The results provide strong evidence that the 4G model leads to improved performance and that this increase in performance is due to the inclusion of global charge equilibration.

A potential limitation of the paper is the extent to which transfer between systems is addressed. The data used here is from Molecular Dynamics trajectories, and the train test splits appear to randomly sample time points along these trajectories. (In some cases, structures are also included from random perturbations to these sampled structures.) Each test structure is thereby likely to be very close to chemical structures included in the training set. This is a fairly limited test of transfer. The results presented here therefore demonstrate that the 4G models provide accurate functional forms for the potentials of specific molecules or periodic systems. For 2G models, past work has found good transfer between molecules within a fairly broad class of molecules. If the authors agree that such transfer between systems remains to be demonstrated for these 4G models, it may be useful to include a brief discussion of this in the paper.

We agree with the reviewer that transferability is an important point to be discussed in the context of ML potentials, which typically have a very flexi-

ble functional form but lack physically motivated terms and constraints. As the reviewer points out, even 2G HDNNPs have show good transferability in previous work although they rely on a local description only. In our present approach we complement this local description by physically meaningful long-range electrostatic interactions, which rely on a globally correct charge distribution. Although the focus of our work has not been on transferability tests but on the presentation and validation of our new method, we are convinced that the inclusion of physical terms will further improve the transferability compared to 2G methods. This expectation is supported by the fact that even traditional charge equilibration schemes with constant electronegativities are known to work well across different systems (Y. Ma, G. K. Lockwood, and S. H. Garofalini, *J. Phys. Chem. C* 115, 24198 (2011)). Furthermore, for the related CENT approach a broad transferability has already been demonstrated for different atomic environments (S. Faraji, S. A. Ghasemi, S. Rostami, R. Rasoulkhani, B. Schaefer, S. Goedecker, and M. Amsler, *Phys. Rev.B*95, 104105 (2017)). We have followed the suggestion by the reviewer and added a paragraph to the manuscript discussing the topic of transferability at the beginning of Section IV (pages 5/6) in the revised manuscript.

The lack of a strong demonstration of transfer between systems may also limit the extent to which this paper is of interest to a broad audience. However, due to the simplicity and elegance of the model, and the clear demonstration that the 4G model can describe multiple charge states, I believe it is of sufficiently broad interest to warrant publication in *Nature Communications*.

We thank the reviewer for this recommendation.

The manuscript is very well written, but I do have two suggestions the authors may want to consider:

1) How are the hardness terms (J of Eq. 2) adjusted during training? In the CENT reference DOI: 10.1103/PhysRevB.92.045131, they are stated as being hand tuned. If these are also hand-tuned here, it may be helpful to include additional details on this process in the supporting information. If they are included in the training of the neural networks, it may be helpful to include this schematically in Fig. 2.

We agree that more information regarding the determination of the hardness will be useful for the reader. The hardness is treated as an element-specific (but not environment-dependent) number that is automatically optimized during the training process like the NN weight parameters. We have now added a sentence on the right side of page 3 to describe this in the manuscript. Since Fig. 2 represents more complicated environment-dependent properties, adding the hardness to the figure would not provide any additional information to the reader. We thus believe that a description in the text only will improve the clarity of the presentation of the method.

2) The results from the 3G model for the Ag clusters are not included in the main manuscript, other than to show the charges in a color scheme attached to the 2G results in Fig. 5. It may be helpful to include the 3G results in the sentence above Fig. 5 that compares the energy and forces of 4G to those of 2G. (In my first reading, I was distracted by the lack of comparison to 3G and why it was not included.)

Indeed the errors of the 3G-HDNNP have not been reported in the main text. We have now added the missing errors for the energies and forces obtained with the 3G-HDNNP at the end of the section on the Ag₃ clusters (page 8) as suggested by the reviewer.

I also noticed a few typos:

First paragraph below Fig. 2: are no physical observables → are not physical observables

First sentence of section III chapter → section

First full paragraph below Table 1: word "the" repeated twice

Consider replacing the phrase "distinguish both clusters" with "distinguish between clusters"

We thank the reviewer for making us aware of these typos, which we have fixed in the revised manuscript.

3 Reviewer #3 (Remarks to the Author):

The authors propose an approach to include non-local information of charge populations into local machine learning models. The approach is elegant, as it uses local quantities predicted by HDNN potentials in combination with a global charge equilibration procedure in order to derive partial charges for an atomistic system of arbitrary charge. These charges are then in turn used to polarize the energy predictions of another set of HDNN potentials. This circumvents several problems local ML approaches have with describing the charge distributions of systems with differently charged states. The advantages of doing so are demonstrated clearly by the conducted experiments. Overall the paper is well written and I would recommend it for publication, provided a few minor issues are addressed.

We thank the reviewer for this recommendation.

The authors should discuss the computational effort of their approach and its implications for practical simulations. Since a system of equations needs to be solved during the charge equilibration, the method formally scales as $O(N^3)$ compared to previous generations of the potential which achieve a scaling of $O(N)$. As such, one would expect that the maximum size of the system which

can be treated or evaluation time during dynamics simulations is reduced.

We agree that the computational performance of the method is of high practical relevance. It is correct that, if simple algorithms are used to solve the system of linear equations, the method scales like N^3 with the number of atoms, in contrast to linear scaling of 2G methods. However, due to the availability of highly optimized algorithms developed for the solution of systems of linear equations (which are omnipresent in science and beyond), extremely fast solutions are possible for small to medium sized systems containing a few thousand atoms, which is currently the typical domain of 2G and 3G-HDNNPs. These algorithms are also frequently found in classical force fields using charge equilibration and long-range electrostatics. For very large systems iterative solvers can be used. For these solvers, the limiting factor of each iteration would be the application of a matrix multiplication involving the matrix A , which corresponds to the calculation of the electrostatic potential arising from the current charge distribution. Numerous low complexity algorithms are known for this kind of problem such as fast multipole methods. In this way it would be possible to reduce the complexity from cubic to linear or nearly linear for very large systems containing tens of thousands of atoms and beyond.

We have followed the suggestion by the reviewer and now included a discussion of computational scaling behaviour including rough timings for a typical system size in the manuscript on pages 4/5.

Regarding the computation of the forces, it should be made clear whether the effect of the positions on the charges are considered in the required derivatives of the energy, as they are now environment dependent. This plays a role in the Coulomb expression for the energy, but also in the standard atomic networks, as these depend on the local charge. E.g.:

$$\frac{dE(G(R), q(R))}{dR_a} = \frac{dE(G(R), q(R))}{dG(R)} \frac{dG(R)}{dR_a} + \frac{dE(G(R), q(R))}{dq(R)} \frac{dq(R)}{dR_a} \quad (1)$$

This could for example influence the force error in Table I of the main text.

Absolutely. These terms, i.e. the derivatives of the charges with respect to positions, are included in the calculation of the forces for both, the electrostatic as well as the short-range part, to ensure that the energies and forces are consistent. The details are given in section 1.3 of the supplementary methods. In the section called “An efficient method for the force computation” we explain how the term $(\frac{dE}{dq} \frac{dq}{dr})$ can be calculated efficiently without having to evaluate the dE/dq part explicitly, which is computationally expensive.

Since the inclusion of these terms is an important aspect of the method, we added some sentences in the manuscript on page 5 to highlight this.

General comments: There is a typo on page 9 in the sentence: “The Al dopant ...”, where “then” is used instead of “than”.

It would be helpful to include braces in the summation in Eq. 2 on page 3 to make clear, that both – the second and third term – are included in the sum over i .

ref 18 could be uploaded to arxiv so that this ref can be found by the reader.

We thank the reviewer for these hints, which we have addressed in the revised manuscript.

Several additional changes have been made to the manuscript, which are highlighted in the revised manuscript:

- We renamed section METHOD to METHODS
- We removed the use of subsection in the RESULTS AND DISCUSSION Section
- We removed Table II and integrated its content into the text.
- We combine Figs. 7 and 8.
- We combined Figs. 10 and 11.
- We added titles to all figures and tables.
- We added a reference to the datasets that we used to train the NNs.
- We have reduced the number of references as requested by Nature Communications.
- Further minor changes have been made to improve the clarity of the text at various places.
- We have replaced the example in Fig. 1b in the introduction by a more illustrative case